# Combination Effects of Integrin-linked Kinase and Abelson Kinase Inhibition on Aberrant Mitosis and Cell Death in Glioblastoma Cells

**DOI:** 10.3390/biology12070906

**Published:** 2023-06-25

**Authors:** Abigail Cunningham, Maddisen Brown, Jonathan Dresselhuis, Nicole Robinson, Keni Hervie, Michael E. Cox, Julia Mills

**Affiliations:** 1Department of Biology, Trinity Western University, Langley, BC V2Y 1Y1, Canada; abigail.cunningham@mytwu.ca (A.C.);; 2Vancouver Prostate Center and Vancouver Coastal Health Research Institute, Vancouver, BC V6T 1Z3, Canadamcox@prostatecentre.com (M.E.C.)

**Keywords:** integrin-linked kinase, Abelson kinase, mitotic arrest, apoptosis, glioblastoma, centrosome declustering

## Abstract

**Simple Summary:**

Glioblastomas are common and aggressive brain tumours that exist in adults and children. Currently, there are no effective treatment strategies. Most cancer cells, including glioblastoma cells, exhibit abnormal centrosomes, the key cellular machinery that controls division. Cancer-causing proteins such as integrin-linked kinase and Abelson kinase occur at abnormal centrosomes and these proteins are chemotherapeutic drug targets. In this study, we use a novel cancer drug combination strategy to target these cancer-causing proteins in vitro. We find that a drug that targets integrin-linked kinase also affects Abelson kinase levels and its localization to the centrosome in dividing glioblastoma cells. Moreover, exposure of glioblastoma cells to chemotherapeutic drugs in combination significantly increased aberrant mitotic division and cell death over individual inhibitors alone. These results indicate that integrin-linked kinase regulates Abelson kinase at mitotic centrosomes and combination drug treatment strategies are more effective than individual inhibitors alone at increasing the lethal mitotic division of dividing glioblastoma cells.

**Abstract:**

In cancer cells, inhibition of integrin-linked kinase (ILK) increases centrosome declustering causing mitotic arrest and cell death. Yet, not all cancer cells are susceptible to anti-ILK treatment alone. We investigate a combination drug strategy targeting ILK and another oncogenic kinase, Abelson kinase (ABL). Drug-concentration viability assays (i.e., MTT assays) indicate that ILK and ABL inhibitors in combination decreased the viability of glioblastoma cells over the ILK drug QLT-0267 alone. Combination strategies also increased aberrant mitoses and cell death over QLT-0267 alone. This was evident from an increase in mitotic arrest, apoptosis and a sub-G1 peak following FAC analysis. In vitro, ILK and ABL localized to the centrosome and the putative ILK kinase domain was important for this localization. Increased levels of cytosolic ABL are associated with its transformative abilities. ILK inhibitor effects on survival correlated with its ability to decrease cytosolic ABL levels and inhibit ABL’s localization to mitotic centrosomes in glioblastoma cells. ILK inhibitor effects on ABL’s centrosomal localization were reversed by the proteasomal inhibitor MG132 (a drug that inhibits ABL degradation). These results indicate that ILK regulates ABL at mitotic centrosomes and that combination treatments targeting ILK and ABL are more effective then QLT-0267 alone at decreasing the survival of dividing glioblastoma cells.

## 1. Introduction

### 1.1. Cancer and Anti-ILK Combination Therapies

The multifunctional proteins integrin-linked kinase (ILK) and Abelson kinase (ABL) are established therapeutic targets for many different cancers with elevated expression of these oncogenic proteins [1,2,3,4]. Among them are a wide variety of solid tumours including glioblastomas and other cancers such as leukemias that are refractory to other treatments [5,6,7,8,9,10]. Individually, ILK and ABL are promising targets for many cancers. However, combination therapies are sought to target resistant cancer cells and optimize therapeutic benefits [11,12,13,14]. The proto-oncogenes ABL and ILK are promising combination drug targets for solid tumours such as glioblastomas as both kinases have a role in their progression. Combination therapies targeting ILK and ABL’s oncogenic counterpart BCR-ABL are preferred for other cancers like leukemia when anti-ABL monotherapies are ineffective [12,13,14]. In fact, genetic and pharmacological inhibition of ILK sensitizes resistant populations of leukemic stem cells to ABL 1 tyrosine kinase inhibitors [13,14]. To date, the anti-mitotic effects of combination therapies that target ILK and ABL at the centrosome have not been explored.

### 1.2. ILK and ABL: Promising Drug Targets at Cancer Centrosomes

The centrosome is also known as the microtubule-organizing centre. It is an organelle that regulates mitotic division and serves as a signalling platform for cancer-causing proteins [15]. Cancer cells have excess numbers of centrosomes known as supernumerary centrosomes [16]. During division, cancer cells can cluster these excess numbers at two poles and divide in a pseudobipolar manner. Emerging evidence indicates that these aberrant or “cancer” centrosomes are promising and selective anti-cancer drug targets [17,18]. Anti-mitotic drugs that target these centrosomes cause them to uncluster and this results in multipolar division and mitotic arrest. Often cells cannot recover from these aberrant mitotic events and they ultimately undergo apoptosis [19] or cellular senescence [20,21].

ILK is a multifunctional protein found in mitotic centrosomes. The ILK inhibitor, QLT-0267, is a promising chemotherapeutic drug as it targets clustered, supernumerary centrosomes found in cancer cells [16]. We have shown that ILK downregulation and ILK inhibition cause unclustering of supernumerary centrosomes during cell division. This results in multipolar division, mitotic arrest and cell death or replicative senescence (an irreversible arrest of cell proliferation and altered cell function) in glioblastoma and retinoblastoma lines [22,23]. Centrosome de-clustering was also induced by downregulating ILK expression with ILK siRNA and this was rescued in by reexpressing FLAG-tagged ILK [22]. During division, cancer cells with supernumerary centrosomes are significantly more sensitive to anti-ILK drugs than normal cells with two centrosomes [16]. Therefore, unclustering supernumerary centrosomes is an “Achilles Heel” of cancer cells and an important mechanism for anti-ILK chemotherapies [16,22,23,24,25,26].

Unlike ILK, ABL’s role in mitotic “cancer” centrosomes is not well understood and most studies have focused on the ABL fusion gene product (BCR-ABL) [27,28]. However, it has recently been shown that mammalian Abl is associated with γ-tubulin at the centrosome in mitotically dividing cells [29]. Moreover, Abl kinase-mediated γ-tubulin phosphorylation appears to be essential for centrosome maturation and integrity [29]. Centrosomal aberrations were observed in Abl and Arg (Abl-related gene, also called Abl2) null cells [29] and in mitotically dividing cancer cell lines exposed to anti-ABL drugs [27,28]. Furthermore, BCR-ABL co-immunoprecipitates with pericentrin (an integral component of the centrosome), and the ABL inhibitor imatinib was shown to decrease this association resulting in centrosome aberrations [30]. The impact of ABL-induced centrosomal changes on the survival of mitotically dividing cancer cells is currently unknown.

The purpose of this study is to further characterize the relationship between ILK and ABL in centrosome function and cell survival of mitotically dividing glioblastoma cells. Exposure of glioblastoma cells to ILK and ABL inhibitors, in combination, increases cytotoxicity and aberrant mitotic events over individual inhibitors alone. We find that ILK and ABL localize to the centrosome and the putative ILK kinase domain is important for ILK’s localization. Furthermore, the effects of the ILK inhibitor on survival correlate with its ability to decrease cytosolic ABL levels and inhibit ABL’s localization to mitotic centrosomes.

## 2. Materials and Methods

### 2.1. Cell Culture, Fractionation and Transfection

Glioblastoma cell lines were plated in dishes or on poly-D-lysine coated coverslips. T98G cultures were grown in DMEM containing 10% serum and 0.1% gentamicin (ThermoFisher, Waltham, MA, USA). U251 and SF188 cells were obtained from Dr. Chinten J. Lim’s laboratory from the Faculty of Medicine, University of British Columbia and were grown in high glucose DMEM containing 10% serum, 1% NEAA and 0.1% gentamicin (ThermoFisher, Waltham, MA, USA).

Fractionation was performed based on the CellLytic NuCLEAR Extraction Kit protocol (Sigma-Aldrich, St. Louis, MO, USA). Cells were washed and harvested in ice-cold PBS before resuspension and incubation in a hypotonic buffer for 15 min. Stock NP-40 solution was added to achieve a final concentration of 0.05% detergent. The cells were immediately vortexed and then centrifuged for 30 s to spin down the crude nuclear fraction. The supernatant (the cytoplasmic fraction) was removed while the nuclear pellet was washed twice with hypotonic buffer. Nuclear extraction buffer was added to the pellet and the pellet was sonicated, vortexed and incubated on ice for 25 min. The nuclear fraction was then removed as the supernatant. The cytoplasmic and nuclear fractions were then immediately used for Western blot analysis or stored at −20 °C short term.

Transfection of DNA constructs was completed using Lipofectamine^®^ 3000 reagent (ThermoFisher, Waltham, MA, USA) according to the manufacturer’s protocol to transfect glioblastoma cells with the ILK-FLAG [22], ABL-FLAG (pcDNA3-Abl-His-Flag, from Addgene) or control DNA construct containing a G418 (ThermoFisher, Waltham, MA, USA) selectable marker. Cells were kept in normal media containing transfection reagent for 3 h before replacement with selection media containing G418.

### 2.2. Drug Exposure

QLT-0267, 25 mM stock in DMSO, Valocor Therapeutics, Vancouver, BC, Cannada.

MG132, 20 mM stock in DMSO, Calbiochem, Imatinib mesylate, Sigma-Aldrich or Stem Cell, Vancouver, BC (solubilized in DMSO according to manufacturer’s instructions). Cells were exposed to drugs (or vehicle control) in complete media containing 10% fetal bovine serum.

### 2.3. Western Blotting

Lysates from glioblastoma cell lines were prepared in HEPES lysis buffer (pH 7.5) containing (in mM) 50 HEPES, 150 NaCL, 1.5 MgCl_2_, 1 EDTA, 1% Triton X-100, 100 NaF, 10 sodium pyrophosphate containing protease and phosphatase inhibitors. Cell pellets were lysed by sonication or resuspension through an 18 g needle followed by vortexing and incubation on ice. Membranes were washed 3× for 10 min with TBS-T between steps. Primary antibodies used include anti-c-Abl (1:500, Cell Signalling) anti-Lamin B1 (1:10,000, Abcam, Cambridge, UK) anti-α-Tubulin (1:500, Sigma-Aldrich), FLAG M1 (1:500, Sigma-Aldrich) as reported elsewhere [22]. Secondary antibodies used included anti-rabbit HRP (1:1000, Cell Signalling) and anti-mouse HRP (1:1000, Cell Signalling). Membranes were incubated with Clarity Western ECL Substrate (Bio-Rad, Mississauga, ON, Canada) for 5 min at room temperature before imaging with a ChemiDoc Imaging System (Bio-Rad, Mississauga, ON, Canada). Membranes were stripped using TBS-T buffer containing β-mercaptoethanol for 30 min at 37 °C prior to reprobing.

### 2.4. Immunocytochemistry and Microscopy

Cells were plated onto PDL-coated coverslips in 6-well plates at 0.4 × 10^6^ cells per well, then left to adhere overnight at 37 °C and 5% CO_2_. Cells were fixed for 3 min in 4% paraformaldehyde following drug exposure. Cells were solubilized using 0.02% Triton-X-100 and then blocked in 5% NGS-BSA-PBS-T. The cells were incubated overnight at 4 °C in primary antibodies for pericentrin (mouse, Abcam Cat# ab28144, 1:200; or, rabbit, Abcam Cat# ab4448, 1:1000), ABL (rabbit, Cell Signalling Technology, 1:100), and α-tubulin (mouse, Sigma-Aldrich, 1:1000) as previously reported [23]. The mouse Abcam pericentrin antibody was used for ABL colocalization studies while the rabbit Abcam pericentrin antibody was used for all other experiments. Following washing, the cells were incubated in secondary antibody, anti-mouse 568 (1:200, Thermo Fisher) or anti-rabbit 488 (1:200, Thermo Fisher). Antibody solutions were diluted in 1% NGS-BSA-PBS-T incubating solution. The cells were washed, then mounted onto glass slides with Vectashield mounting media containing DAPI (Vector Laboratories).

The Olympus IX81 inverted microscope with a disk scanning unit (DSU) spinning disk confocal was used to image immunocytochemistry slides. Images were further analyzed using MetaMorph Premier and ImageJ software. Calculated total cell fluorescence (CTCF) was used to analyze the intensity of fluorescence in ImageJ. CTCF was measured as the difference between the integrated density and the product of the selected cell area and the mean fluorescence of the background.

## 3. Results

### 3.1. Combination Anti-ILK and Anti-ABL Drugs Were More Efficacious at Inhibiting Survival Than Individual Drugs Alone

ABL and ILK are highly expressed in glioblastomas and both have been individual chemotherapeutic targets for this tumour model [2,3,5,6]. Glioblastoma cell viability following drug exposure was examined to determine whether chemotherapeutic drugs, targeting ILK and ABL in combination, were more efficacious than individual drugs alone. QLT-0267 has been previously described [11,31] as a specific inhibitor of ILK [32,33] and imatinib, is an FDA-approved ABL1 tyrosine kinase inhibitor that targets the constitutive tyrosine kinase activity of BCR-ABL in chronic myelogenous leukemia. Imatinib also inhibits the related mammalian tyrosine kinase ABL [34,35]. Survival effects were examined in various glioblastoma cell lines (Figure 1A–C). SF188 cells were exposed to the ILK inhibitor, QLT-0267 and the ABL inhibitor, imatinib (IM) alone or in combination for 96 h. An MTT assay was performed immediately following the drug exposure to quantify the viability of the cultures. The MTT assay quantifies viable cells using a colorimetric change as the yellow, soluble MTT substrate is reduced by metabolically active cells to formazan, which is solubilized in DMSO, creating a purple solution that is analyzed using a spectrophotometer [36]. A strong correlation between cell viability and formazan production has been reported [37,38]. A marked decrease in the viability of SF188 cells was observed when cells were exposed to an anti-ILK drug QLT-0267 (10 µM) or nilotinib (3, 5 and 10 µM) for 96 h. This was decreased further when QLT-0267 was used in combination with the ABL inhibitors imatinib (10 µM) or nilotinib (3, 5 and 10 µM) (Figure 1A). Likewise, the effects of QLT-0267 were examined alone or in combination with the ABL drugs imatinib or nilotinib in T98G and U251 cells. In T98G cells, QLT-0267 exerted a significant decrease in viability as did 5 and 10 µM imatinib or 3, 5 and 10 nilotinib alone (Figure 1B). This was further decreased when the ILK inhibitor was used in combination with ABL inhibitors at all concentrations (Figure 1B). In U251 cells, another commonly used experimental glioblastoma model, a 96-h exposure to 10 µM QLT-0267 resulted in a significant albeit less robust decrease in viability as compared to control. As with T98G glioblastoma cells, combination drug exposures using QLT-0267 with either imatinib or nilotinib resulted in a significant decrease in viability (as compared to QLT-0267 alone).

In past studies, we found that anti-ILK therapies in vitro increased apoptosis in retinoblastoma and glioblastoma cell lines [22,23], as measured using propidium iodide labelling and FACS analysis. Indeed, we observed an increase in the number of cells in sub-G1 (indicative of apoptotic cells) when T98G cells were exposed to 10 µM QLT-0267 for 72 h as compared to vehicle control. This effect was further increased when cells were incubated with 10 µM QLT-0267 together with 10 µM imatinib (Figure 2A). At a shorter drug exposure time of 24 h, although QLT-0267 alone was not significantly different from the control, there was a marked increase in apoptotic cells in the QLT-0267 and nilotinib combination treatment groups for all concentrations of nilotinib (10 µM QLT-0267 with 3, 5 or 10 µM nilotinib; Figure 2B). Combination treatment effects on apoptosis also correlated with a significant decrease in the average number of cells in the field of view (expressed as % C; Figure 2C).

### 3.2. ILK and ABL Inhibitors in Combination Increase Aberrant Mitoses over Individual Inhibitors Alone

ILK and ABL have both been studied individually for their role in cancer cell division and survival [2,3,5,6]. Inhibition of ILK [16,22,23,24] or the gene fusion product, BCR-ABL [27,28], has resulted in aberrant mitosis and decreased cell survival while ABL inhibition compromised the mitotic spindle in cell lines [29]. The role of ABL in cancer centrosomes has not been studied and the effects of drugs that target ILK and ABL at cancer centrosomes is unknown. Here, we investigated the effects of combining ILK and ABL inhibitors on mitosis. T98G cells were exposed to QLT-0267 or imatinib (IM) alone or in combination for 6 h. A comparison of early mitotic changes was undertaken to determine whether combination drug treatments resulted in a greater number of aberrant mitoses and mitotically arrested cells than individual drug treatments. Following drug exposures, glioblastoma cells were immunostained for α-tubulin and pericentrin and counterstained with DAPI. Spindle poles were identified by immunolabelling centrosomes and spindles with an anti-pericentrin and anti-α-tubulin antibody, respectively. The percentage of mitotic cells (as a percentage of total cell number) was then quantified in these unsynchronized cell populations. Mitotic arrest was evident by the fact that cells exhibited an increase in the aberrant mitotic division while overall cell numbers decreased (Figure 3A). In keeping with past studies, QLT-0267 increased the proportion of cells exhibiting mitotic arrest at 10 µM, relative to vehicle control (Figure 3A). Unlike QLT-0267, imatinib alone did not significantly increase the number of mitotically arrested cells (Figure 3A). QLT-0267 and imatinib in combination induced a significant increase in mitotically arrested cells above individual drug treatments (Figure 3A). Values were averaged and expressed as percent control ± SEM. *p* < 0.05, * different from control; ** different from QLT alone, as determined by analysis of variance (ANOVA) and Fisher’s (LSD) test.

In T98G cells, multipolar spindles, disorganized mitotic spindles and pericentrin fragmentation were observed in the presence of 10 μM QLT alone and these effects were exacerbated in cells treated with QLT together with 5 and 10 μM imatinib (Figure 3B). Imatinib alone did not result in significant changes to mitoses. Likewise, T98G cells were treated with 10 μM QLT-0267 (QLT) alone or together with an alternate ABL inhibitor, nilotinib (Nil) at 5 or 10 μM in combination. Mitoses appeared even more aberrant in these combination treatment groups (i.e., number of poles and disorganized spindles) as compared to QLT alone. Additionally, a pronounced increase in apoptotic cells was seen in these combination treatment groups (Figure 3C), *n* = 3–4.

### 3.3. Wild-Type FLAG-Tagged ILK and ABL Constructs Localize to Mitotic Centrosome

ILK localizes to the centrosome of cancer cells [16,22,23,24] and we have shown that ILK immunoreactivity was significantly inhibited by ILK siRNA treatment [16,22,23,24]. To date, the importance of the putative ILK kinase domain in trafficking ILK to the centrosome has not been studied directly. Here, we expressed wild-type (WT) and kinase-dead (KD) FLAG-tagged ILK constructs in glioblastoma and immunostained cells for pericentrin and FLAG. ILK-WT colocalized with pericentrin while ILK-KD did not (*n* = 3; Figure 4A). Our findings indicate that ILK’s putative kinase domain regulates its trafficking to the centrosome. This would be predicted to alter its interaction with other microtubule-regulating proteins located here [16].

ABL kinase has recently been shown to phosphorylate and promote γ-tubulin ring complexes located at the microtubule organizing centre and facilitate the pericentriolar material formation and centrosome maturation [29]. To study ABL’s localization to centrosomes in glioblastoma cancer cells, FLAG-tagged ABL or an empty control construct was expressed in glioblastoma cells that were immunolabelled for pericentrin and FLAG. FLAG-tagged ABL colocalized with pericentrin immunostaining while no centrosomal immunostaining of the empty construct was not observed in controls (Figure 4B), *n* = 3 representative of separate platings. Given that the ILK interactome and ABL are found at the centrosome and share common protein partners [16,29], we sought to examine the effects of ILK inhibition on ABL localization.

### 3.4. Inhibition of ILK Decreases ABL Levels in the Cytosol and at Mitotic Centrosomes

When ABL’s oncogenic counterpart BCR-ABL localizes to the cytosol, proliferation increases and increased cytosolic ABL has been associated with more aggressive cancers [39]. Whether or not ABL’s centrosomal localization is affected by ILK is unknown. We first examined the amount of ABL in subcellular fractions and the effects of the ILK inhibitor QLT-0267 on ABL’s subcellular location. Western blots of subcellular fractions were performed following a 6 h drug exposure. The purity of fractions was verified in Western blots that were run in parallel and probed for proteins that are enriched in nuclear (lamin B1) or cytoplasmic (α-tubulin) fractions. Levels of ABL were much higher in the cytosolic fraction of glioblastoma cells than in the nuclear fraction and exposure of the cells to QLT-0267 significantly decreased cytosolic ABL (Figure 5). ABL has been shown to be regulated by the ubiquitin-dependent proteasomal pathway. To determine whether ILK’s regulation of ABL occurred in a proteasomal-dependent manner MG132, a proteasomal inhibitor was used in combination with QLT-0267. MG132 rescued the ILK-dependent decrease in cytoplasmic ABL levels as compared to the vehicle control (Figure 5A). Densitometric analysis of proteins on Western blots was averaged across trials and expressed as a percentage of the DMSO drug vehicle control ± SEM, *n* = 5, *p*-value < 0.05 as determined by analysis of variance and Tukey’s multiple comparison test (Figure 5B).

ILK is found at the centrosome of a variety of cancer cells where it regulates several pathways related to division and mitotic spindle formation [16,22,23,24]. Furthermore, ABL has recently been found to occur at centrosomes during mitotic division in nontransformed cells [29]. Therefore, we investigated whether ABL, a protein whose levels are tightly regulated by the proteasome, is also found at centrosomes in dividing glioblastoma cancer cell lines and if its centrosomal localization is altered by inhibitors of either ILK or proteasomal degradation. T98G cells were exposed to 10 µM QLT-0267 (QLT) ± 1 or 10 µM MG132 (1 and 10 MG) for 6 h (Figure 6). Cells were then immunocytochemically stained using an ABL and pericentrin (PCNT) antibody to observe whether drugs induced a change in ABL levels at the centrosome. ABL levels were averaged across trials and calculated as the amount of ABL immunofluorescence at the centrosomes above the cellular background (Figure 6B). In drug vehicle controls, immunocytochemical labelling indicates that ABL is found at the centrosomes and co-localizes with pericentrin (Figure 6A,B). ILK inhibition resulted in a loss of ABL immunoreactivity from mitotic centrosomes (Figure 6A,B) and this was rescued in the presence of the proteasomal inhibitor MG132. These results indicate that ABL’s localization to the mitotic centrosome is regulated by ILK activity and that inhibition of ABL’s proteasomal degradation rescues this effect. Although the proteasomal inhibitor MG132 did not change the nuclear portion of ABL kinase, MG132 did change the nuclear intensity and localization of ABL kinase at mitotic centrosomes. As cells were not synchronized, this discrepancy is likely because centrosomal ABL at mitotic centrosomes represents a small portion of overall ABL in mitotically dividing and nondividing glioblastoma cells. Whether ABL is a part of the ILK interactome at centrosomes is currently unknown. Relatedly, future research is needed to determine how the ILK protein complex regulates ABL function.

## 4. Discussion

The centrosome serves as a scaffold for many signalling proteins in cancer cells including ILK [16,24] and ABL (Wang et al., 2022). ILK exists as a tubulin-based multiprotein complex at centrosomes where it acts to cluster supernumerary centrosomes found in cancer cells [16,24]. This is thought to be an adaptive cancer cell mechanism resulting in pseudobipolar mitosis and cell survival. ILK performs its centrosome clustering activity in a focal adhesion-independent but centrosome-dependent manner through ILK’s centrosome-interacting partners TACC3 and ch-TOG [16]. The TACC3-ch-TOG complex clusters supernumerary centrosomes in cancer cells in an ILK-and Aurora-A-dependent manner [16]. QLT-0267- and siRNA-mediated downregulation of ILK block TACC3′s association with Aurora A [24,26]. A TACC3 phosphorylation site is required for centrosome clustering and ILK was shown to regulate this phosphorylation [16]. Examination of mitotic spindle poles in control and ILK siRNA-treated cells revealed a depletion of ILK at mitotic spindle poles in retinoblastoma and Hela cells [22,24]. However, the importance of the putative ILK kinase domain in trafficking ILK to the centrosome has not been studied. Here, we extend these findings as we observed a marked decrease in the co-labelling of FLAG-tagged ILK-KD with pericentrin relative to FLAG-tagged ILK-WT. ABL has also been found to colocalize with pericentrin and γ-tubulin at the centrosome in cancer cells expressing GFP- and Myc-tagged ABL constructs [29]. ABL phosphorylates γ-tubulin at the γ-tubulin ring complex, regulates its nucleation function at centrosomes and facilitates PCM formation and centrosome maturation [29]. We find that ABL also localizes to the centrosome in glioblastoma cells (Figure 4 and Figure 6). Moreover, inhibition of ILK decreased ABL here (Figure 6). Thus, ILK and ABL inhibitors together, may work in a complementary manner to inhibit centrosome function in cancer cells. Indeed, increased cytotoxicity of ILK and ABL inhibitors in combination (Figure 2) correlates with aberrant mitotic events in glioblastoma cells (Figure 3). Real-time microscopy will help determine whether cancer cells undergoing aberrant mitotic events are being selectively lost. Future work is needed to unravel the mechanism underlying this combination drug response.

Members of the ILK interactome may also alter the proteasomal degradation of ABL indirectly through phosphorylation. ABL is activated through phosphorylation [40] and shortly after ABL activation, it is negatively regulated by phosphotyrosine phosphatases [41] and irreversibly downregulated by ubiquitin-dependent proteasomal degradation [42]. In this way, proteasomal systems can attenuate kinase signalling by coupling kinase activation with subsequent protein degradation [43]. Activated and tyrosine phosphorylated forms of ABL are more unstable than wild-type and kinase-inactive forms of ABL [40]. Increased ABL tyrosine phosphorylation would be predicted to increase its activity and its rate of proteasomal degradation [42]. This is because phosphorylated ABL is more rapidly degraded [40,44]. ILK is also ubiquitinated at a minimum of 14 sites [45] and ubiquitylation of ILK is involved in the regulation of its degradation through the proteasomal pathway. The ILK interactome may regulate ABL degradation at centrosomes as these organelles have been shown to coordinate the local degradation of proteasomal substrates [42]. This is supported by our results, indicating that an ILK inhibitor decreased ABL levels at the centrosome in mitotic cells and this was rescued by the proteasomal inhibitor MG132 (Figure 5). Given the importance of the proteasome in the regulation of centrosomal and cell cycle proteins [46] future studies are needed to examine this potential crosstalk between ILK and ABL regulation at the level of the proteasome.

ILK affects several downstream signalling pathways, yet the method of this regulation is still controversial. ILK was originally identified as a serine/threonine kinase [47]. Since then, ILK has been shown in hundreds of publications to promote phosphorylation of proteins and kinases and highly purified recombinant ILK has been unequivocally demonstrated to have protein kinase activity [48,49]. Moreover, small molecule inhibitors have been identified that bind specifically to the putative kinase domain [49]. On the other hand, ambiguities over ILK’s catalytic capacity arise because it lacks several important motifs that are conserved in most kinases (reviewed in [50,51]). This is supported by extensive genetic studies in flies, worms and mice, showing that the putative kinase activity is not required for its function [50,51]. Nevertheless, many other atypical eukaryotic protein kinases are bona fide kinase targets having potent inhibitors with proven therapeutic value (reviewed in [49]). Despite the controversy, the continued demonstration of ILK as a therapeutic target, alone or in combination with other chemotherapeutics, lends credence to the development of potent compounds to inhibit ILK’s oncogenic activity.

Regardless of the mechanism, several recent studies have shown that ILK downregulation or inhibition sensitizes cancer cells to various chemotherapeutics that target ABL or BCR-ABL. For example, an unbiased genome-wide CRISPR-Cas 9 screen in triple-negative breast cancer revealed that loss of ILK and its binding partners α-parvin and PINCH-1 potentiated the inhibitory effect of bosutinib, a SRC/ABL kinase inhibitor [52]. Similarly, ILK knockout cells were more sensitive to SRC/ABL kinase inhibition and more likely to undergo G1 arrest and apoptosis [52]. Additionally, an emerging role for ILK-mediated signalling is especially promising in chronic myeloid leukemia (CML), a disease caused by the constitutively active tyrosine kinase BCR-ABL. ILK is upregulated in (CML) progenitor and leukemic stem cells especially in patients resistant to the tyrosine kinase inhibitors (TKI) imatinib and dasatinib [14]. Genetic and pharmacological inhibition of ILK was shown to sensitize TKI-nonresponsive leukemic stem cells to ABL TKI [14]. Moreover, ILK and TKI used in combination show strong synergistic effects on cancer cells while having no adverse effects on normal bone marrow cells [14]. Like leukemic stem cells, our study indicates that combination therapies that target ABL and ILK in glioblastoma may optimize therapeutic benefits for these drug-resistant and aggressive CNS cancers. Combination therapies are sought to develop alternative therapeutic strategies for a wide variety of solid tumours, such as gliomas in situations where there is resistance to ABL or ILK inhibitors alone. These in vitro drug targets will need to be validated using in vivo models or organoid models [53] of patient-derived xenografts as a critical step to clinical translation.

## 5. Conclusions

In this study, we examined the relationship between ILK and ABL at mitotic centrosomes in glioblastoma cell lines. A relationship between ILK and ABL has not been exclusively studied until now even though they: (1) share common binding partners; (2) have similar cellular localization; and (3) regulate common signalling pathways and cellular processes including centrosome function [1,26,29,44,45,54]. We find that the ILK inhibitor QLT-0267 acts together with ABL inhibitors to decrease cycling and viability and that the enhanced cytotoxicity of ILK and ABL inhibitors in combination correlate with their impact on the mitotic arrest. Finally, we find that ILK inhibition decreased ABL levels at the centrosome and this correlates with aberrant mitotic events and survival effects in glioblastoma. Centrosome clustering occurs via phosphorylation of the microtubule regulating protein TACC3, a protein in the ILK interactome [16]. Given that this and other proteins at the centrosome are also targetable for cancer [16], future studies will need to explore how these proteins interact with ILK and ABL to regulate centrosome function and mitoses in cancer cells.

## Figures and Tables

**Figure 1 biology-12-00906-f001:**
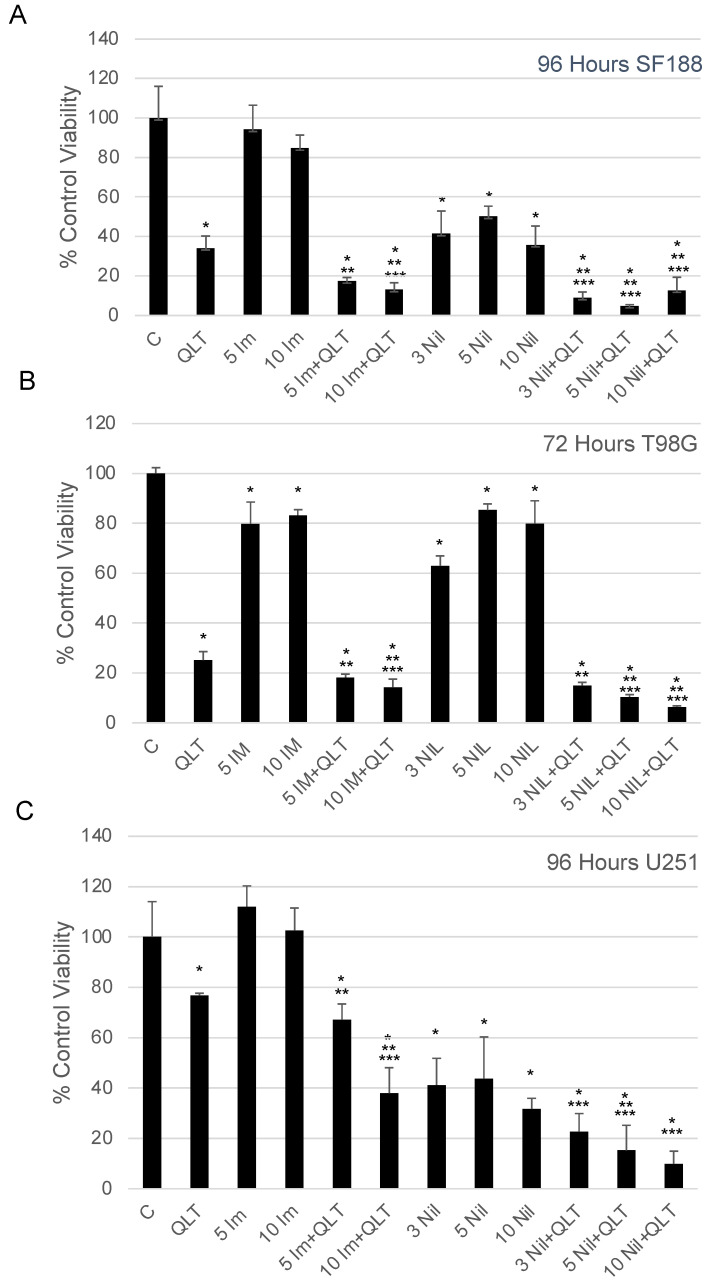
ILK and ABL inhibitors in combination significantly decrease viability over individual inhibitors alone. (**A**) An MTT viability assay was performed after SF188 cells were exposed to QLT-0267 (10 μM) with or without imatinib (5 and 10 μM) or nilotinib (3, 5 and 10 μM) for 96 h. *n* = 5 separate platings, with each treatment group run in triplicate or quadruplicate and averaged. * different from vehicle C, ** different from ABL inhibitor at the same concentration and *** different from QLT alone; determined using an ANOVA and Fisher’s (LSD) test. *p* < 0.05. (**B**) An MTT viability assay was performed after T98G cells were exposed to QLT-0267 (10 μM) with or without either imatinib (5 and 10 μM) or nilotinib (3, 5 and 10 μM) for 72 h. *n* = 3–4 separate platings. * different from vehicle C, ** different from ABL inhibitor at the same concentration; determined using an ANOVA and Fisher’s (LSD) test. *** different from QLT alone; determined using a paired *t*-test. *p* < 0.05. (**C**) An MTT viability assay was performed after U251 cells were exposed to QLT-0267 (10 μM) with or without either imatinib (5 and 10 μM) or nilotinib (3, 5 and 10 μM) for 96 h. *n* = 3 separate platings. * different from vehicle C, ** different from ABL inhibitor at the same concentration and *** different from QLT alone; determined using an ANOVA and Fisher’s (LSD) test. *p* < 0.05.

**Figure 2 biology-12-00906-f002:**
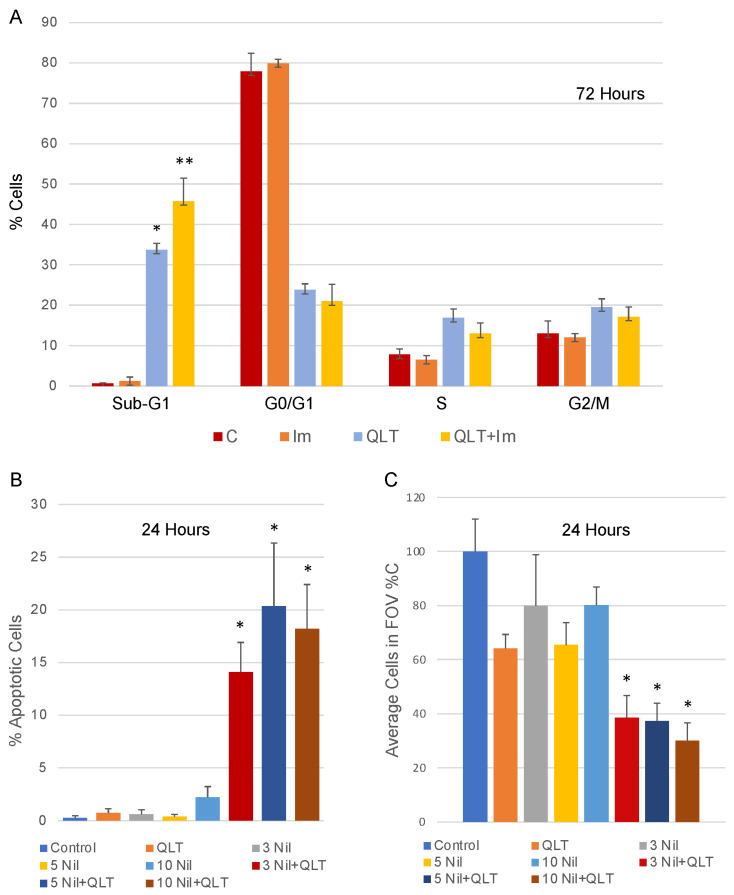
ILK and ABL inhibitors in combination significantly increase G1/S cell cycle arrest and apoptosis over individual inhibitors alone. (**A**) FACS analysis was performed in T98G cells treated with QLT-0267 (10 μM) with or without imatinib (5 and 10 μM) for 72 h. Nonadherent and adherent cells were pooled. The percentage of cells in sub-G1, G0/G1-, S- and G2/M is expressed and an average ± SEM. * different from vehicle C; ** different from QLT alone. *n* = 3 separate platings. ANOVA and Fisher’s (LSD) test. *p* < 0.05. (**B**) Apoptotic cells were quantitated in T98G cells treated with QLT-0267 (10 μM) with or without imatinib (5 and 10 μM) or nilotinib (3, 5 and 10 μM) for 24 h. (**C**) Likewise, the average cells in the field of view (represented as a percentage of control) are shown. * different from C, QLT alone or nilotinib alone at the same concentration. *n* = 3–4. ANOVA and Fisher’s (LSD) test. *p* < 0.05.

**Figure 3 biology-12-00906-f003:**
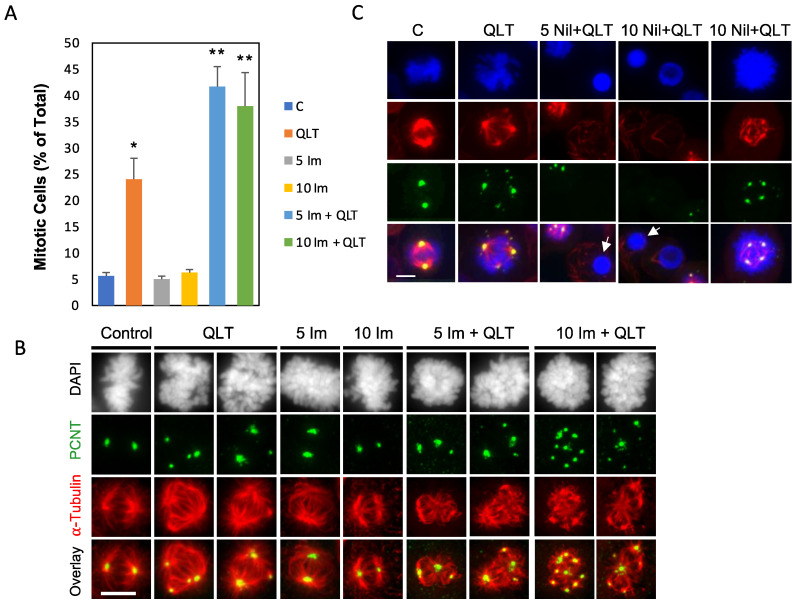
ILK and ABL inhibitors in combination exert a synergistic effect on aberrant mitoses and mitotic arrest over individual inhibitors alone. Mitotic cells were quantitated in T98G cells after exposure to the following drugs: 10 μM QLT-0267 (QLT) and 5 or 10 μM imatinib alone (5 Im or 10 Im) or in combination and compared to drug vehicle control (Control). Cells were exposed to drug(s) or drug vehicle in complete media containing serum for 24 h, fixed and immunocytochemically stained with an α-tubulin and pericentrin antibody. Nuclei were counterstained with DAPI. (**A**) The percentage of mitotic cells was determined for each treatment group; 5–10 fields of view were randomly chosen and analyzed per trial. The total cell number analyzed is over 1800. QLT-0267 significantly increased the percentage of mitotic cells above vehicle control while imatinib was not different from control. QLT-0267 together with imatinib was significantly greater than QLT alone. Values were averaged and expressed as percent control ± SEM. *p* < 0.05, * different from control; ** different from QLT alone, *n* = 4. Significance was determined by an ANOVA and Fisher’s (LSD) test. (**B**) Representative T98G cells following exposure to 10 μM QLT-0267 (QLT) and 5 or 10 μM imatinib alone (5 Im or 10 Im) or in combination with QLT-0267. Multipolar spindles, aberrant mitotic spindles and pericentrin fragmentation were observed in the presence of QLT alone or together with imatinib. Scale bar = 10 µm. (**C**) Representative T98G cells following exposure to 10 μM QLT-0267 (QLT) alone or together with 5 or 10 μM nilotinib in combination. Aberrant mitotic events (i.e., multipolar spindles and pericentrin fragmentation) occur in the presence of QLT alone or together with nilotinib. Additionally, a pronounced increase in apoptotic cells (labelled with white arrows) was seen in combination treatment groups. *n* = 3–4. Scale bar = 10 µm.

**Figure 4 biology-12-00906-f004:**
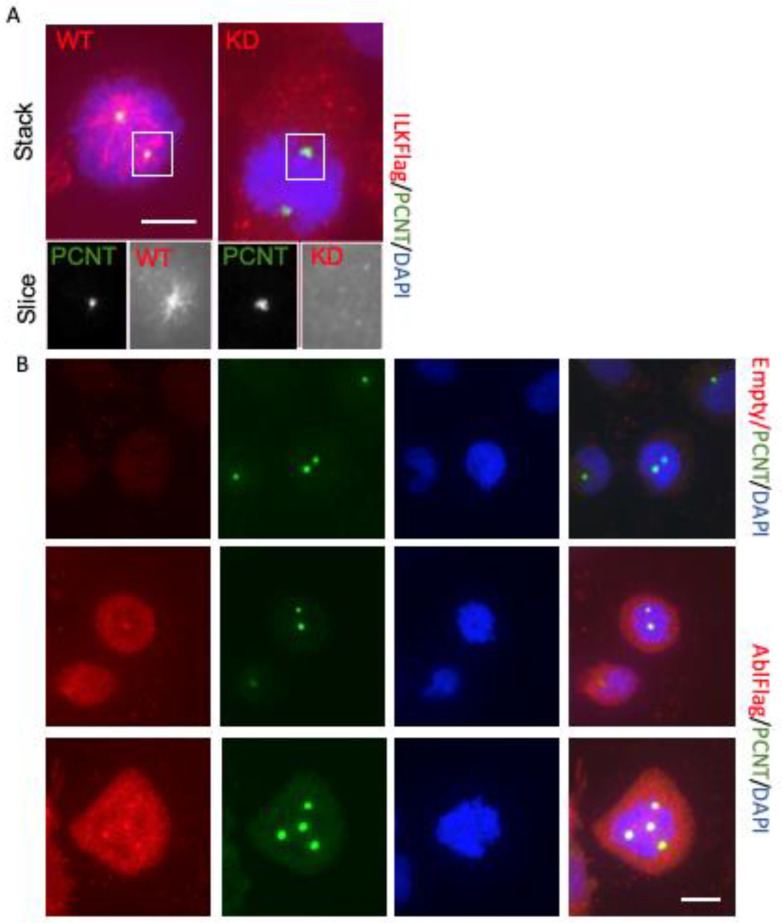
ILK and ABL localize to the centrosome in glioblastoma cells. (**A**) In T98G cells, FLAG-tagged ILK-WT colocalized with pericentrin immunostaining while the ILK mutant, FLAG-tagged ILK-KD did not. (**B**) FLAG-tagged ABL was seen to colocalize with pericentrin while the empty control plasmid did not. Shown are representative figures of three separate platings. Large multinucleated cells undergoing cell division were frequently seen (as shown in the lower panel of the FLAG-tagged ABL tranfected cells).

**Figure 5 biology-12-00906-f005:**
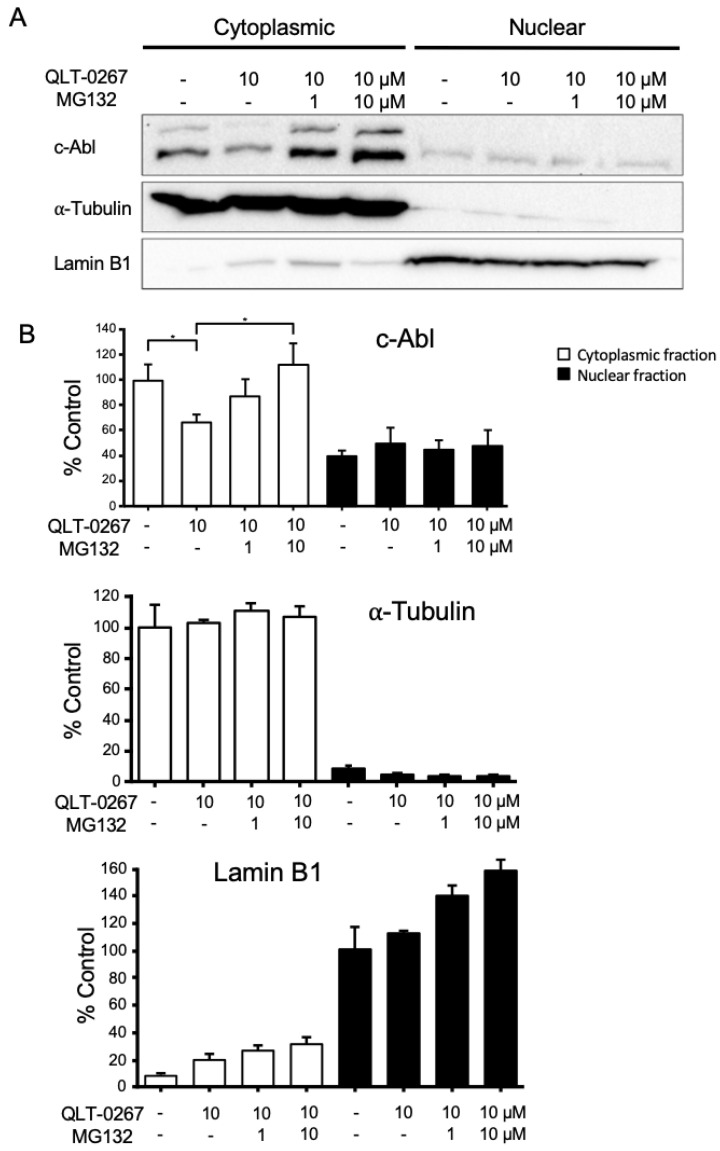
The ILK inhibitor QLT-0267 decreases ABL in the cytosolic fraction and the proteasome inhibitor MG132 rescues these effects in T98G cells. (**A**) T98G glioblastoma cells were exposed to either DMSO or 10 μM QLT-0267 (QLT) alone or together with 1 μM or 10 μM MG132 for 6 h. Western blot analysis following cell fractionation indicates that increasing amounts of MG132 in the presence of QLT-0267 rescued cytoplasmic ABL levels. The purity of cellular fractions was verified using the primarily cytosolic (α-tubulin) and nuclear (Lamin B1) proteins. Please check the original images in Appendix A. (**B**) Following Western blotting, densitometric analysis of proteins was performed. Values were averaged and expressed as percent control ± SEM. *p* < 0.05, * different from control or QLT-0267 together with 10 μM MG132 as determined by a paired Student’s *t*-test, *n* = 5 independent trials.

**Figure 6 biology-12-00906-f006:**
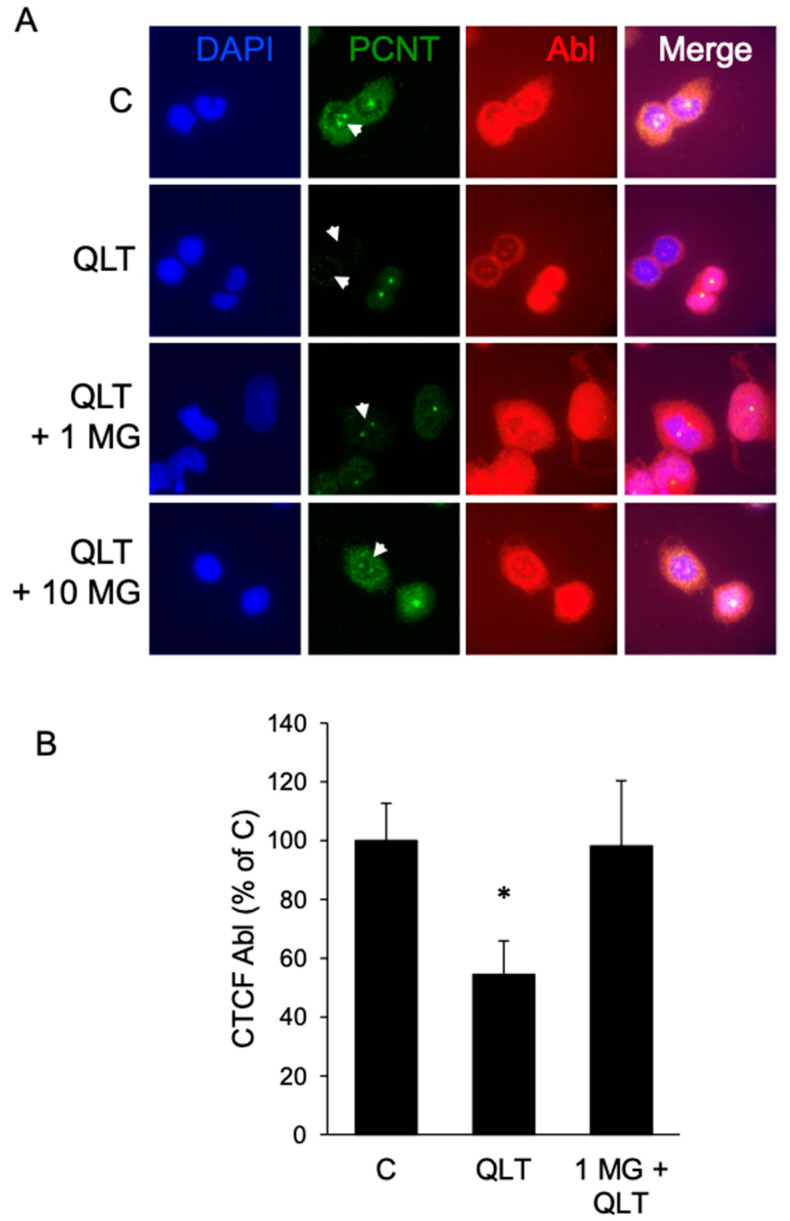
The ILK inhibitor QLT-0267 decreases ABL at the centrosome in glioblastoma cells. T98G cells were exposed to drug vehicle, 10 μM QLT-0267 alone or together with increasing concentrations of MG132 (1 μM or 10 μM) for 6 h in serum-containing media. Cells were then immunocytochemically labelled using an ABL and pericentrin antibody and nuclei were counterstained with DAPI. (**A**) Loss of ABL immunofluorescence at the centrosome of mitotically dividing cells was observed in T98G cells exposed to 10 μM QLT-0267 and this loss was rescued by 1 μM and 10 μM of the proteosomal inhibitor MG132. White arrows highlight mitotically dividing cells. An overlay of mitotically dividing cells is shown at low (see “Overlay” above) and high magnification (see “Mag” above). Calibration bars represent 10 μm. (**B**) ABL immunofluorescence was quantified at mitotic centrosomes in glioblastoma cells. The calculated total cell fluorescence at the centrosome (CTCF) was determined for C, 10 μM QLT-0267 alone or together with 1 μM MG132. Data represent *n* = 5 (separate culture platings) for T98G cells. *p* < 0.05, * different from control.

## Data Availability

Data contained within the article are available on request from the corresponding author.

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
