# Peer review of "Combination Effects of Integrin-linked Kinase and Abelson Kinase Inhibition on Aberrant Mitosis and Cell Death in Glioblastoma Cells"

_biology, 2023, doi:10.3390/biology12070906_

Round 1

Reviewer 1 Report

This manuscript investigates the effects of combining inhibitors to two proteins that are widely accepted to be key to cancer cell growth: ILK and Abl. The study is performed in cell culture models of glioblastoma, a cancer with clear un-met clinical needs. It is an interesting study that has yielded some novel results, particularly the ILK inhibitor decreasing cytoplasmic levels of Abl. However, as it stands, some of the key conclusions are not supported by the data, as outlined under “major comments” below. Specific comments on some of the figures are also provided.

Major Comments:

1.       “Synergistic Effects” of ILK and Abl inhibitors are mentioned in the title and throughout the paper, including in figure 2 and 3 figure legend titles and in introduction, results and discussion text.

However, a simple definition of synergy in terms of drug interactions is that the total effect is greater than the sum of the individual effects of each drug, i.e. that the drugs are acting more than additively. There is no evidence provided that the effects of the ILK and Abl drugs are acting synergistically.  Therefore, either evidence should be provided this is the case, or suitable alternative terms should be used instead.

2.       Evidence that ILK inhibition is decreasing centrosomal Abl is not strong. The western blot in Figure 5 does show a decrease in cytosolic Abl upon QLT treatment. However, the immunofluorescence images in figure 6 are at low resolution and do not support decreased centrosomal Abl. They do support a decrease in cytoplasmic/total Abl but if anything it looks like centrosomal localisation of Abl is more prominent following QLT treatment.

As both of these points are key to the paper as it is, they would need to be properly addressed.

Specific comments on figures:

Fig1B: Legend: Significance stated for p<1.0 in this figure. Perhaps this is a typo?

Figure 1B. Combination of both drugs does produce a larger effect than either alone, but no evidence shown to suggest they are synergistic.

Fig2a: Graph needs labels on X and Y axes.

Fig3: Legend: Significance stated for p<0.10 in this figure. This is highly unusual in biology. Perhaps this is a typo? Or if p<0.10 is being used, a justification for this must be provided.

Fig 3 legend: Scale bar units should be µm, not µM.

Figures 4 and 6: No scale bars on images.

Fig6A: Images not high enough resolution to see if Abl localised to centrosome or not.

Arrows on images are on pericentrin, but not Abl channel. In contrast to what is stated in text, it looks like QLT decreases total/cytoplasmic Abl, but not centrosoaml Abl, as red puncta still clearly visible in QLT image.

Higher resolution imaging, with more careful use of arrows and/or zoomed inserts of centrosomes required so valid conclusions can be drawn.

6B: Legend says “ABL immunofluorescence was quantified at mitotic centrosomes in glioblastoma cells. The calculated total cell fluorescence (CTCF)……..”

What was quantified, centrosomal fluorescence or total cell fluorescence?

Author Response

Letter to Reviewer 1:

We would first like to thank Reviewer 1 for their suggestions and for having caught a labelling error for Fig. 6.  With these revisions I am confident that the manuscript has been made significantly stronger.  A copy of the manuscript with track-changes has been included.

Major Comments:

  1. We have changed the title of our manuscript in light of this reviewer’s comment on “synergy in terms of drug reactions”. The revised title of the manuscript is “Combination Effects of Integrin-Linked Kinase and Abelson Kinase Inhibition on Aberrant Mitosis and Cell Death in Glioblastoma Cells.
  2. We apologize for the mislabelling of Figure 6. Pericentrin and ABL labels were mistakenly switched. Also the figure resolution was low. This led to some confusion. We have revised the labels, increased the resolution and added a panel of figures at higher magnification.

Specific Comment on Figures:

1) Fig. 1B, including the significance (revised to read p<0.5) and the word “synergistic” to describe the effects of drug combinations has now been revised.

2) Fig. 2B labels have been included on X and Y axes.

3) Changes to Fig. 3 legend have been made such that p<0.05.  The scale bar typo has been corrected.

4) Fig. 4 and 6 now have scale bars.

5) Fig. 6A is represented with higher resolution.  Furthermore, more careful use of arrows and higher magnifications of mitotic cells has been added.

Reviewer 2 Report

The manuscript by Cunningham A., et al reported the synergistic effect of integrin-linked kinase and Abelson kinase inhibitors on inducing mitotic arrest and cell death in glioblastoma cells. They also showed that ILK and ABL proteins localized to mitotic centrosome, and inhibition of ILK decreased ABL levels in the cytosol but not in the nucleus.

Overall, many data are conflict with each other, eg. Fig. 1A vs 1B, Abelson kinase inhibitor imatinib (IM) alone significantly reduced cell viability in 1B, but not in 1A. Fig. 1A vs Fig. 2A, integrin-linked kinase inhibitor QLT significantly reduced cell viability in 1A, but in 2A, QLT treatment did not affect cell cycle even though the bar of QLT was similar to the QLT+IM. Fig. 2A vs Fig. 3A, QLT treatment did not affect cell cycle in 2A, QLT treatment increase the mitotic cell population in 3A. Fig 3C, the morphology of PCNT was abnormal in Nil+QLT lanes. Fig. 4B, bottom lane, the PNCT morphology is different to the empty control. Fig. 5A vs Fig. 6A, Fig. 5A showed that the proteasome inhibitor MG132 did not change the nuclear portion of Abl kinase, but Fig. 6B showed that MG132 changed the nuclear intensity and localization of Abl kinase.

Other points,

Drug Nil is not provided in materials.

FLAG tag is not consistent.

Line 152, 4 °C.

Many Figure labels are not consistent, such as Nil, NIL,  hrs, hours.

Line 289.   This title does not make sense.

Figure 5B, normalization of Abl to Tubulin for cytosolic fraction and to Lamin B1 for nuclear fraction is needed.

Author Response

We would first like to thank Reviewer 2 for their suggestions, for with these changes I am confident that the manuscript has been made significantly stronger.

Reviewer 2 reported a conflict with Fig. 1A and 1B regarding imatinib’s effect on cell viability.  This may have been because: 1) imatinib was used from two different sources (see Methods) and 2) the inherent variability of this data set.  To avoid confusion, we have removed Fig. 1A.  We have replaced it with additional data using a third glioblastoma cell line SF188.

Reporting of Fig. 2A led to some confusion.  Specifically, in Fig. 2A QLT-0267 alone did indeed alter the cell cycle but we merely reported that QLT-0267 and imatinib was significantly different from QLT and Control.  We have now changed this to read “* different from vehicle C; ** different from QLT alone” (line 195 of revised).  This change will help to resolve the apparent discrepancy between Fig. 2A and Fig. 3A that was reported by Reviewer 2. 

This reviewer commented on the abnormal morphology of pericentrin in lanes 3 and 4.  This is indeed true as there are apoptotic cells in the field of view.  We have endeavoured to make this clearer by describing these figures in greater detail in the figure legend.  We have distinguished between mitotic and apoptotic cells with arrows.  Also, Fig. 4B now includes a better description of a large (and likely multinucleated) cell undergoing a multipolar division. 

Regarding the apparent discrepancy between Fig. 5A and 6A, the calculated total cell fluorescence at the centrosome was rescued by MG132.  However, overall nuclear Abl was not changed by MG132.  As cells were not synchronized, this discrepancy is likely because centrosomal Abl at mitotic centrosomes represents a small portion of overall Abl in mitotically dividing and nondividing glioblastoma cells.  This is also include in the revised manuscript (see line 301). 

Other Points:

1) We have now included “Nil” in the materials.

2) Flag-tagged ABL has been corrected in the body of the manuscript.

3) 4°C has been corrected

4) Figure labels (i.e., Nil, NIL, hrs) should now be consistent.

5) Line 289 in the online version that I received did not have a title.  Therefore, I am not sure which title this reviewer is referring to.

6) Figure 5B – normalization.  This data has been archived by a former student and I will need more time to gain access to this and to do these calculations.

Reviewer 3 Report

Cunningham et al. provided data that ILK and ABL inhibitors synergize on causing aberrant mitosis and cell death in GBM cells. This is an interesting finding, and the data is generally well presented. The manuscript should be strengthened along these following lines before publication in Biology.

Major points:

1.     The authors used serum-dependent adherent cultures of glioblastoma cell lines for all experiments, while recent evidence suggests that serum-free cultures are better representations of glioblastoma biology. The authors should repeat key experiments in serum-free gliosphere cultures or laminin-coated adherent cultures. Also, the authors mostly relied on a single cell line T98G for all experiments. They should repeat key experiments in other GBM cell lines.

2.     While the authors use the term “synergistic” to describe the relationship between ILK and ABL inhibitors in different phenotypes, they don’t explicitly test whether the effects they observe is truly synergistic or additive. The authors should perform statistical tests to see if the effects are truly synergistic.  

3.     The authors exclusively used sub-2N DNA as a marker for apoptosis. They should verify that the cells are indeed apoptotic (as opposed to other forms of cell death) by staining with more specific apoptosis markers, e.g. cleaved caspase.

4.     While the authors used two different concentrations of ABLi throughout the manuscript, they used a single concentration of ILKi. To better establish the dose dependence and the concentration regime where synergy exists between ILKi and ABLi, the authors should perform concentration titration for both ILKi and ABLi simultaneously.

5.     The authors observed aberrant mitoses when treating cells with ILK or ABL inhibitors. While they claim on line 263 that there were more aberrant mitotic events in ILKi+ABLi conditions, it is difficult to tell from the limited number of sample images whether this is indeed true. The authors should quantify aberrant mitotic events and present statistics between different treatment groups.

6.     Similarly, the authors should quantify FLAG-ILK co-localization with pericentrin in Figure 4A instead of relying on a single sample image. Also, it appears from the single sample image in Figure 4A that the expression of KD-ILK is much lower than that of WT-ILK. Is the failure of KD-ILK to localize to the centrosome an artifact of poor expression? The authors should demonstrate that KD-ILK and WT-ILK are expressed at similar levels in the cell by Western blotting.

7.     Does ILKi treatment affect the localization of ILK to the centrosome? What about ABLi treatment and the localization of ABL to the centrosome? This seems like an obvious mechanism for the proposed synergy of ILKi and ABLi in causing mitotic defects.

Minor points:

1.     In Figure 2A, the authors should label the bar plot groups sub-G1, G0/G1, S, G2/M to make it easier for readers to compare the bar plots.

2.     Scale bars should be included for all microscopy images, e.g. Figures 4 & 6.

Author Response

We would first like to thank Reviewer 3 for their suggestions for with these changes we are confident that the manuscript has been made significantly stronger.  We have addressed Major and Minor points below.

Major Points:

  1. This author has suggested that we use gliosphere cultures or laminin-coated adherent cultures. While these are good suggestions, I am required to submit revisions within 10 days and cannot do these revisions within this time frame.  However, Reviewer 3 did request that we repeat key experiments in other GBM cell lines and to this end we have added another MTT assay using SF188 cells (see revised Figure 1).

  1. Regarding a “synergistic” relationship, we have changed the title of the manuscript to: “Combination Effects of Integrin-Linked Kinase and Abelson Kinase Inhibition on Aberrant Mitosis and Cell Death in Glioblastoma Cells.” Additionally, the word “synergistic” to describe the effects of drug combinations in the body of the manuscript have now been revised.

  1. This reviewer indicated that we exclusively used sub-2N DNA as a marker for apoptosis. However, we did observe and quantitate apoptotic cells based on nuclear fragmentation (see Fig. 2B and Fig. 3C). The suggestion that we stain for more specific apoptotic markers is valid however, we cannot perform this within a 10-day period (the time that we have been given to make these revisions).

  1. Concentration titrations will not be required to establish synergy given changes that we have made (see #2 above).

  1. This reviewer is correct in saying that it is difficult to tell from the limited number of samples whether there are more aberrant mitotic events in combination drug treatment groups. Moreover, when longer exposure times are used, we observe decreased numbers of cells in the field of view that correlates with multipolar division and an increase in apoptosis (see Fig. 2).  However, without real-time microscopy we cannot determine whether cells undergoing an aberrant mitotic event are being selectively lost.  We have included this in the discussion (see line 335 of the revised manuscript).

  1. Quantitation of FLAG-ILK colocalization with pericentrin and Western blotting to control for KD-ILK and WT-ILK expression levels cannot be done in 10 days along (along with the revisions that have been requested by other reviewers).

  1. This reviewer indicated that the ILK and Abl inhibitor effects on Abl centrosomal localization should be examined. One confound is that the commercial ILK antibody that we have used in the past for ILK’s centrosomal localization (Sikkema et al., 2014) is no longer available.  Given this confound, these experiments can’t be done in a 10 day period.

Reviewer 4 Report

Comments 

  • The section provides a clear and concise summary of the study's objectives and findings.
  • However, it would be helpful to provide more context and background information on glioblastomas and their current treatment strategies for readers who may not be familiar with the topic.
  • Additionally, it would be useful to provide a brief explanation of the significance of the study's findings and how they may contribute to the development of more effective treatment strategies for glioblastomas.

  • The abstract provides a comprehensive summary of the study, including the methodology, results, and implications of the findings.
  • The use of technical terms and abbreviations may make it difficult for readers who are not familiar with the subject matter to understand the content. It would be helpful to include brief definitions or explanations of technical terms and abbreviations.
  • The abstract could benefit from a more concise and clearer writing style. Some sentences are long and complex, making it challenging for readers to understand the main message.

  • The section provides a comprehensive overview of the topic and explains the significance of ILK and ABL as potential drug targets in cancer cells.
  • The use of technical terms and abbreviations may be challenging for readers who are not familiar with the topic. It would be helpful to include brief definitions or explanations of technical terms and abbreviations to improve readability.
  • The section provides a good introduction to the study's objectives and sets the stage for the discussion of the study's findings.
  • However, some sentences are long and complex, making it difficult to follow the main message. It would be helpful to use shorter and more concise sentences where possible to improve clarity.
  • Overall, the section provides a good introduction to the study's focus on the use of ILK and ABL as drug targets and sets the stage for the discussion of the study's findings.

Material and Methods

  • In the cell culture section (lines 102-110), it would be useful to specify the passage number of the cells used in the experiments, as this can affect their behavior and response to treatments.
  • In the drug exposure section (lines 128-133), it would be helpful to include the final concentration of each drug used in the experiments.
  • In the immunocytochemistry and microscopy section (lines 148-159), it would be useful to describe how the cells were counterstained with DAPI and whether this was done before or after mounting the slides with Vectashield.
  • In the Western blotting section (lines 134-147), it would be helpful to provide more information about the protease and phosphatase inhibitors used in the lysis buffer, as different inhibitors can have different specificities and potencies.
  • Finally, it may be useful to include a table or figure summarizing the antibodies used in the experiments, their sources, and dilutions. This would make it easier for readers to quickly reference this information.

Results

The authors investigated the efficacy of combining ILK and ABL inhibitors to inhibit glioblastoma cell survival. They found that the combination of QLT-0267 and imatinib or nilotinib was more effective at decreasing cell viability and inducing apoptosis than individual drugs alone. Furthermore, they observed that the combination of ILK and ABL inhibitors resulted in a significant increase in mitotically arrested cells with aberrant mitosis. They also demonstrated that ILK and ABL constructs localize to the centrosome and that inhibition of ILK decreased ABL levels in the cytosol and at mitotic centrosomes. The authors' findings suggest that targeting both ILK and ABL may be an effective therapeutic strategy for glioblastoma treatment.

3.2 This section provides a clear and detailed description of the experimental design and results of the study investigating the effects of ILK and ABL inhibitors on mitosis in glioblastoma cell lines. The authors provide sufficient information to understand the rationale for the study, the methods used, and the results obtained.

One potential area for improvement is in the clarity of the language used. For example, in sentence 240, the phrase "the effects of combination therapy on mitosis" could be revised to "the effects of combining ILK and ABL inhibitors on mitosis." Additionally, in sentence 244, it might be helpful to clarify that the cells were immunostained for α-tubulin and pericentrin in order to visualize the spindle poles, rather than to identify the cells as mitotic.

The authors could also benefit from providing more context for their findings, such as discussing how their results compare to previous studies investigating the effects of ILK and ABL inhibitors on cancer cell division and survival. Finally, it would be useful for the authors to clearly state the implications of their findings and how they contribute to our understanding of the role of ILK and ABL in glioblastoma.

3.3 This section provides a clear and concise description of the experimental design and results of the study investigating the localization of ILK and ABL constructs to the mitotic centrosome. The authors present their findings in a straightforward and easy-to-understand manner.

One potential area for improvement is in the organization of the information presented. For example, the authors could benefit from grouping the information about ILK and ABL constructs separately, rather than interspersing the two throughout the section. Additionally, it would be helpful for the authors to provide more context for their findings, such as discussing how their results relate to previous studies investigating the localization of ILK and ABL in cancer cells.

Finally, the authors could benefit from more clearly stating the implications of their findings and how they contribute to our understanding of the role of ILK and ABL in glioblastoma. Additionally, it would be useful for the authors to provide some potential hypotheses or directions for future research based on their findings.

3.4 Detailed and well-organized description of the experimental design and results of the study investigating the effects of ILK inhibition on ABL levels in the cytosol and at the mitotic centrosome. The authors present their findings in a clear and logical manner, making it easy to understand the rationale for the study, the methods used, and the results obtained.

One potential area for improvement is in the clarity of the language used. For example, in sentence 309, the phrase "BCR-ABL within the cytosol" could be revised to "the oncogenic counterpart of ABL, BCR-ABL, has been shown to localize within the cytosol." Additionally, it might be helpful for the authors to provide more context for their findings, such as discussing how their results relate to previous studies investigating the regulation of ABL levels in cancer cells.

Finally, the authors could benefit from more clearly stating the implications of their findings and how they contribute to our understanding of the role of ILK and ABL in glioblastoma. Additionally, it would be useful for the authors to provide some potential hypotheses or directions for future research based on their findings, such as investigating the downstream pathways regulated by ABL and ILK in glioblastoma cells.

Discussion

It’s well-written and effectively summarizes the main findings of the study. It provides a thorough discussion of the implications of the results and offers suggestions for future research directions.

One suggestion for improvement is to provide more detailed explanations for some of the technical terms used in the discussion. For example, it may be helpful to define what is meant by "proteosomal degradation" and provide more background on how this process works.

Additionally, while the discussion section does touch on the potential clinical implications of the study, it would be helpful to provide more specific details about how these findings could be applied in the development of new treatments for cancer. For example, the authors could discuss the potential for ILK and ABL inhibitors to be used in combination therapy and how this approach could be further investigated in preclinical and clinical trials.

Overall, the discussion section effectively highlights the significance of the study's findings and provides valuable insights for future research directions.

Conclusions

Provides a clear summary of the study's findings and their implications. It effectively highlights the main results of the study, including the relationship between ILK and ABL at mitotic centrosomes in glioblastoma cells and the synergistic cytotoxicity of ILK and ABL inhibitors.

One suggestion for improvement would be to include a brief discussion of the limitations of the study, such as potential confounding factors or limitations in the methodology. This would help provide a more balanced interpretation of the results and highlight areas for future research. Additionally, it would be helpful to provide more specific suggestions for future studies, such as which proteins at the centrosome could be targeted for cancer therapy and how they may interact with ILK and ABL.

Author Response

We would like to thank Reviewer 4 for providing very clear instructions on suggested changes to the manuscript.  These have greatly improved the clarity of this paper.

As requested, we have provided more background and context on glioblastomas and current treatment strategies in the conclusion.  The abstract has been rewritten to be more concise.  We have shortened long and complex sentences throughout the manuscript.  We have also included brief explanations of technical terms (i.e., FACS and MTT assays).

In the Material and Methods we have addressed the following:

1) Passage number of cell lines. 

2) Final drug concentrations.

3) Counterstaining with DAPI (specifically the use of Vectasheild mounting media containing DAPI).

4) A list of protease and phosphatase inhibitors used in the Western blotting section.

5) A revision of sentences (lines 240 and 244) as suggested.

6) A clarification of why immunostaining for alpha-tubulin and pericentrin was used (i.e., to visualize spindle poles) has been added.  Line 209, “Spindle poles were identified by immunolabelling centrosomes and spindles with an anti-pericentrin and anti-alpha-tubulin antibody, respectively”.

7) More context for effects of ILK and ABL inhibitors on cancer cell division and survival and how these results compare with previous studies has been provided.

As per this Reviewer’s request, we have separated information about ILK and ABL.  We have also provided context for our findings stating:  “Given that ILK and ABL are both found at the centrosome and share common protein partners we sought to examine the effects of ILK inhibition on ABL localization” (line 257 of revised manuscript).  Later we state that “Our findings indicate ILK’s putative kinase domain also regulates its trafficking to the centrosome.  This would be predicted to alter its interaction with other microtubule regulating proteins located here”.

3.4 I have changed the sentence (line 309) as instructed by this reviewer.  I have also provided more context to this finding stating that “When the oncogenic counterpart of ABL, BCR-ABL localizes to the cytosol, proliferation increases and increased cytosolic ABL is associated with more aggressive cancers.” (line 268 of revised manuscript).

This reviewer requested that we provide some potential hypotheses or directions for future research …. Therefore, we have added: “Whether ABL is a part of the ILK interactome at centrosomes is currently unknown (line 304 of revised manuscript).  Relatedly, future research is needed to determine how the ILK protein complex regulates ABL function”. (line 305)

Discussion

Additional information about proteosomal degradation is provided in the discussion (lines 343-344).

This reviewer requested that the authors could discuss the potential for ILK and ABL inhibitors to be used in combination therapy and how this approach could be further investigated in preclinical and clinical trials.  Therefore we have added an additional line: “These in vitro drug targets will need to be validated …..” (line 378)

Conclusions

Reviewer 4 suggested that we provide specific suggestions for future studies.  Therefore, we have added an additional sentence to do so (see line 376).

Round 2

Reviewer 1 Report

I thank authors for responding to my comments and believe the manuscript is now significantly improved.

Author Response

We would like to thank Reviewer 1 for their very helpful review and understand from the comments and checklist below that we have met this Reviewer's requirements for submission.

Reviewer 2 Report

Thank you for addressing my questions.

The staining of PCNT in Fig. 6 looks abnormal and different to Fig. 4.

Some unrelated red wavy lines are seen in Fig. 4 and 6.

Unrelated labels: “Chart title” in Figure 1, “…value:23.8” in Fig 2.

Peri in Figure 4 needs to be changed to PCNT.

Author Response

We would like to thank Reviewer 2 for catching some typographical errors in the figures and for pointing out immunostaining differences between figures.

Reviewer 2 mentioned that pericentrin labelling in Fig. 6 looked different from pericentrin staining in Fig. 4.  This is because in Fig. 6 we used a different antibody for pericentrin (a mouse AbCam antibody, ab28144) then the one used for the other figures (a rabbit AbCam antibody, ab4448).  Immunolabelling albeit weaker for this antibody (as was evident even in by the manufacturer’s representative figure) did label pericentrin nonetheless. We have included the Cat. #’s for these different antibodies in Section 2.4 of Materials and Methods (line 136).  We have also included the following description: “The mouse Abcam pericentrin antibody was used for ABL colocalization studies while the rabbit Abcam pericentrin antibody was used for all other experiments” (line 137 and 138).

Other Minor Changes:

2) We have removed the wavy lines in Fig. 4 and 6 under some of the labels

3) We have removed the unrelated labels “chart title” in Fig. 1 and “value 23.8” in Fig. 2

4) We have changed “peri” in Figure 4 to “PCNT” for consistency

Reviewer 3 Report

The authors have not experimentally addressed most of the points brought up by me or the other reviewers. I understand that the authors were only given ten days for revisions. I suggest that the editors give authors more time to perform revision experiments as suggested by the reviewers. If this is not possible, I suggest the authors submit to another journal with a more flexible revision time frame.

Author Response

Reviewer 3 states that “we have not experimentally addressed most of the points brought up by me or the other reviewers”.  However, only one other reviewer has suggested minor changes in this second review and we have now addressed these.  Reviewer 3 has made some suggestions to use a different research paradigm (i.e., gliosphere cultures or laminin-coated adherent cultures).  These are interesting experiments but we would need several months to perform these given that some of these experiments would require further investigation and our current lack of expertise.  For example, if we were to perform laminin studies we would also need to investigate the integrin receptor subtype that is involved.  Alternatively, if we used 3D cultures, reproducibility of results (a common problem for 3D cultures in general) might be problematic especially as we would be using this technique for the first time.

Round 3

Reviewer 2 Report

Thank you for addressing all my questions. I have no further questions.